# Controlling the Progression of Curvature in Children and Adolescent Idiopathic Scoliosis Following the Administration of Melatonin, Calcium, and Vitamin D

**DOI:** 10.3390/children9050758

**Published:** 2022-05-21

**Authors:** Alexandru Herdea, Mihai-Codrut Dragomirescu, Alexandru Ulici, Claudiu N. Lungu, Adham Charkaoui

**Affiliations:** 111th Department of Pediatric Orthopedics, Carol Davila University of Medicine and Pharmacy, 050474 Bucharest, Romania; alexherdea@yahoo.com; 2Pediatric Orthopedics Department, Grigore Alexandrescu Children’s Emergency Hospital, 011743 Bucharest, Romania; mcodrutd@gmail.com; 3Department of Surgery, Clinical Country Emergency Hospital Galati, 800578 Galati, Romania; lunguclaudiu5555@gmail.com; 4Department of Morphological and Functional Sciences, Faculty of Medicine and Pharmacy, Dunărea de Jos University of Galați, 800008 Galați, Romania; charkaoui.adham@gmail.com

**Keywords:** idiopathic scoliosis, quality of life, vitamin D, melatonin, calcium

## Abstract

Idiopathic scoliosis affects a severe number of children. Their quality of life and development are also disturbed. Some therapeutic strategies have been developed to control illness progression and to optimize the quality of life. In this perspective, randomized, case-control, interventional study, the impact of using melatonin, calcium, and vitamin D, respectively, on idiopathic scoliosis patients was analyzed. Our preliminary results showed that these drugs positively affected the illness progression quantified by the spine curvature. Patients with idiopathic scoliosis may benefit from a novel treatment by supplementation with vitamin D, calcium, and melatonin.

## 1. Introduction

Scoliosis—a term derived from ancient Greek, where “scoliosis” translates as crooked, bent—is an evolving spine disease that consists of lateral deviation in the frontal plane, necessarily associated with vertebral rotation [1,2]. The lack of vertebral rotation along with lateral deviation denotes scoliotic attitude, a benign deformity that disappears in supine [1,2]. Vertebral deformity with a Cobb angle greater than 10 degrees in the frontal plane defines scoliosis, and any value under this threshold is called spinal asymmetry [3,4].

Scoliosis is a three-dimensional spinal deformity that affects between 2% and 4% of the pediatric population [5,6]. The most affected age group is adolescence, accounting for 90% of cases diagnosed in children. Girls are more commonly affected by scoliosis, with a ratio of 1.5–3:1 to boys [7,8].

The causes of scoliosis are known in about 20% of cases—neurological, congenital, syndromic—with the remaining 80% falling within the definition of idiopathic scoliosis [9,10]. Although the etiology is unknown, many factors have been considered related to the occurrence and evolution of scoliosis. Genetic conditioning, vestibular dysfunction, alterations in muscle and connective tissue, endocrine disorders, and mineral metabolism have been proposed as causes of idiopathic scoliosis [11,12].

A causal relationship has been established between melatonin deficiency and scoliosis in fish, laboratory mice, and also in humans [13]. Melatonin is a hormone synthesized by the pineal gland which acts on the sleep–wake rhythm, as well as on free radical detoxification. It has an antioxidant effect, it is involved in bone formation and protection, and it participates in the proper functioning of the reproductive, cardiovascular, and immune systems [14]. In bone metabolism, melatonin has a stimulating effect on osteoblast differentiation and type I collagen synthesis, and an inhibitory effect on bone resorption by inhibiting RANKL-mediated osteoclast formation and activation [15,16]. Low melatonin levels have been linked to idiopathic scoliosis [17]. Serum levels vary in a circadian manner, but for the standardization of our study, we established the sampling at 8:00 a.m. and 9:00 a.m. The role of vitamin D3 in phosphocalcic metabolism is well known. Its average serum level is between 30 ng/mL and 100 ng/mL. Deficiency is problematic in both children—rickets—and adults—osteomalacia—and correlates with an increased incidence of fractures due to bone fragility and with the prospect of associated muscle damage [18]. One study found that 91% of patients with scoliosis were deficient in vitamin D3 [18]. A meta-analysis of the relationship between phosphocalcic balance and the pathogenesis of idiopathic scoliosis recommends testing patients for serum vitamin D3 [19]. Studies show that the administration of vitamin D3 and calcium has a protective role in patients with idiopathic scoliosis by slowing down the progression of the Cobb angle [20,21].

The treatment of adolescent idiopathic scoliosis is staged according to Stagnara’s scheme [22]. Patients with a Cobb angle below 30 degrees receive physical therapy, the recommended method being Schroth [23]. Those with an angle between 30 and 50 degrees receive the recommendation to wear a spinal brace and to perform physical therapy. Severe cases with a Cobb angle of over 50 degrees are candidates for surgical treatment [24]. Thus, scoliosis treatment is pathogenic, not etiological, focusing only on the effects of biological and molecular mechanisms that are not fully known. More recently, researchers have looked into custom braces to improve respiratory parameters and quality of life [25].

Given the critical role that vitamin D, calcium, and melatonin may play in both the onset and the development of adolescent idiopathic scoliosis, this novel study aims to verify whether the administration of melatonin, vitamin D, and calcium supplements to patients diagnosed with idiopathic scoliosis, having a Risser score of 0–3, will slow down or stop the progression of the curvatures.

Drug therapy for idiopathic scoliosis is a relatively new trend in dealing with such a complex pathology. Although the interest in such studies is increasing, few publications are noticed [26,27]. The perspective of such works is quite attractive, impacting both the patient’s quality of life and the overall treatment cost of such patients. Another aspect noted is that calcium, melatonin, and vitamin D are available worldwide, including in developing countries where idiopathic scoliosis is also present. Furthermore, these drugs can be easily administered without requiring particular protocols and specialized personnel.

## 2. Materials and Methods

The study took place in the Pediatric Orthopedics Department at “Grigore Alexandrescu” Children’s Emergency Hospital, Bucharest, Romania, located in an urban area, between 2017 and 2020. The ethics committee of “Grigore Alexandrescu” Children’s Emergency Clinical Hospital of Bucharest approved this study on 10 October 2017. The identification number of the survey is 12/10 October 2017. Informed consent was obtained from the parents of all the participants.

The medical center has been treating patients with scoliosis since the 1980s, having surgeons with experience in diagnosing and treating scoliosis in children and adolescents.

A prospective, randomized, case–control, interventional study, in the period 2017–2020, was carried out on a group of 51 patients aged between 7 and 16 years, selected according to the criteria of inclusion and exclusion among children who presented to the clinical ward of Pediatric Orthopedics or the outpatient clinic, as well as among patients in the records of the physical medicine and balneology wards. Patients diagnosed with idiopathic scoliosis during the study period were selected, and patients who were already diagnosed before the study period with this disease met the inclusion criteria.

Odd and even numbers randomized the patients. Every odd-numbered patient went into the study group, while the even-numbered patient went into the control group.

The study was not blinded for the researchers due to the need to monitor the treatment plan.

Inclusion criteria were: a positive diagnosis of idiopathic scoliosis with a Cobb angle greater than 10 degrees, age between 7 and 16 years, a follow-up period of at least one year from the start of the study.

Exclusion criteria were: scoliosis secondary to malformations (exp. hemivertebra, vertebral blocks), scoliosis belonging to other syndromes or diseases with spinal components, neurological scoliosis, patients older than 16 years and younger than seven years, muscular dystrophy, recent history of vitamin D, calcium or melatonin supplementation, a follow-up period of less than one year, lack of blood tests, low adherence to treatment and established follow-up periods, patient’s desire to be excluded from any study phase.

The working protocol was as follows: the patients were examined clinically and radiologically to confirm idiopathic scoliosis diagnosis. Cobb angle was measured by digital radiography using a high-precision digital system.

Patients who met the inclusion criteria were asked to regularly perform the following blood tests according to Figure 1: 25-OH-vitamin D, total calcium, melatonin.

For vitamin D, the values for 25-OH-vitamin D were considered as follows: toxic value: over 100 ng/mL, normal value: between 30 ng/mL and 100 ng/mL, poor value: between 20 ng/mL and 29 ng/mL, and insufficient value: less than 20 ng/mL. For calcium, the values of total calcium were considered as follows: hypercalcemia: over 10.6 mg/dL, normal value: between 8.80 mg/dL and 10.6 mg/dL. For melatonin, the values were as follows: for the blood test performed at 8:00 a.m.: standard between 8 and 16 pg/mL, and for the blood test performed at 9:00 a.m.: standard between 3 and 8 pg/mL.

To have minimal variations, analyses of 25-OH-vitamin D, calcium and melatonin were performed in the same laboratory. To have the most negligible variations in melatonin in the group of patients, all participants completed the blood test at 08:00 a.m.

The tests were performed only after the prescription and the doctor’s recommendation, collaborating with the study’s primary investigators.

Following the diagnosis of idiopathic scoliosis, the patients received treatment according to the current treatment criteria (Stagnara criteria]: 0–30 degrees: physiotherapy, 30–50 degrees: physiotherapy + brace (Cheneau brace), over 50 degrees: surgery.

Patients underwent either conventional physical therapy or Schroth therapy and were monitored every six months for follow-up.

Patients have been divided into four groups: Risser score 0–1: Cobb angle < 30° and Cobb angle > 30°; Risser score 2–3: Cobb angle < 30° and Cobb angle > 30 °, respectively. 

The following therapeutic interventions were applied to the study group: oral treatment: patients would receive melatonin (1.5 mg/day) at night before bed, vitamin D3 (2000 I.U./day) in the first part of the day, calcium (600 mg/day) in the first part of the day; recovery treatment—physiotherapy (classic or Schroth therapy); a Cobb angle of over 30 ° would also involve wearing a brace.

Patients were monitored every six months for progression of the Cobb angle and reassessment of post-study measurements. The study group repeated the blood samples every six months (except for melatonin which was repeated every 12 months).

The control group performed blood tests at the beginning of the study and was monitored for the progression of the Cobb angle.

The study duration was at least one year for each patient included.

Demographics, Cobb angle, serum vitamin D, calcium, and melatonin were interpreted statistically. *T*-test, F-test, and ANOVA were used to observe the association between vitamin D, melatonin, age, and Cobb angle. Statistical significance was considered for a *p* less than 0.05 with a 95% confidence interval. Data analysis was performed with Microsoft Excel 2016 and Statistical Package for the Social Sciences (SPSS) version 18.

Additionally, other associations were assessed: distribution of patients by age, sex, Cobb angle, serum melatonin, vitamin D, and calcium; evaluation of the response to treatment according to the type of physiotherapy performed, Lenke type, patient age, Risser score, Cobb angle, patient sex, serum level of melatonin, vitamin D, and calcium and administration of supplements.

The statistical power computed for a sample size of *n* = 51 was 0.9383857.

Vitamin D (25-OH-vitamin D3) was analyzed by Architect i2000 (II), chemiluminescent microparticle immunoassay (CMIA). Total calcium was analyzed by the Beckman Coulter AU680 spectrophotometric method. Melatonin (MELA) was investigated by the LCMS method.

## 3. Results

A total of 51 patients met the required criteria, of which 26 were in the study group and 25 in the control group. In addition, patients had a follow-up period of at least one year. Figure 2 shows the distribution of patients by age and sex. Of the 51 patients, 44 were girls and seven were boys.

A comparison between the study group and the control group is shown in Table 1**.**

Analysis of the initial Cobb angle between the study group and the control group is shown in Table 2. The two groups had equal variances (*p* = 0.13 > 0.05) and had no statistically significant differences (*p* = 0.35 > 0.05). SD = standard deviation, CI = confidence interval.

The evolution of the Cobb angle in both groups is shown in Table 2. For the control group, the average initial Cobb angle was significantly lower than the average Cobb angle after one year, which meant an evolution of the disease (*p* = 6.81 × 10^−6^). For the study group, the average initial Cobb angles did not differ significantly from the mean Cobb angles after one year, which would be equivalent to a stagnation of the disease (*p* = 0.24). 

Analysis of the Cobb angle at one year of treatment in both the study group and the control group is illustrated in Table 2. The two groups had different variances (*p* = 0.01), which showed that the average Cobb angles after one year were significantly lower in the study group than in the control group (*p* = 0.005).

The treatment applied to the patients from the study group was assessed, as seen in Table 3.

Analysis of the evolution of the Cobb angle for patients who initially had vitamin D3 deficiency grouped by their final serum levels can be seen in Table 3. The two patient samples showed no statistically significant variances (*p* = 0.35). The results were statistically significant (*p* = 6.85 × 10^−5^), showing an improvement for those who managed to correct their vitamin D deficiency.

Analysis of the evolution of the Cobb angle is shown in Table 4, measuring the impact of each type of physiotherapy that was done.

Initial Cobb angle analysis for patients who underwent classical physiotherapy vs. Schroth physiotherapy is illustrated in Table 4. The two groups did not have significantly different variances (*p* = 0.12). The initial Cobb angle did not differ significantly (*p* = 0.96).

Analysis of the evolution of the Cobb angle after one year of treatment for patients who underwent classical physiotherapy vs. Schroth physiotherapy is shown in Table 4. The two groups did not have significantly different variances (*p* = 0.10) and the results regarding the Cobb angle after one year of treatment did not change significantly (*p* = 0.45).

Analysis of the initial Cobb angle by age group is shown in Table 5. The two age groups had equal variances (*p* = 0.48) and we concluded that the averages of the initial Cobb angles were significantly lower for the first age group than for the second one (*p* = 0.0004).

Analysis of the Cobb angle evolution by age groups is illustrated in Table 5. There were substantial differences between the two age groups (*p* = 0.009), but no significant differences in the evolution of the Cobb angle concerning the age group (*p* = 0.40).

The evolution of the Cobb angle one year after treatment by age groups is discussed next in Table 5. There were no significant differences between the variances of the two age groups (*p* = 0.32), but the average of the absolute Cobb angles was considerably lower in the first age category than in the second (*p* = 0.002). However, the differences between the Cobb angle averages in the same category were not statistically significant.

The analysis of the relationship between age group, vitamin D3 deficiency, and the initial Cobb angle, respectively, showed us that only the age factor had a statistically significant impact (*p* = 0.0009 < 0.05) on the initial Cobb angle.

The analysis of the relationship between the type of physical therapy vs. control or study group and the evolution of the Cobb angle was performed and showed that the interaction between the two variables was not relevant to the evolution of the Cobb angle, in ANOVA analysis.

The analysis of the interactions between the age group, the batch type, and the evolution of the Cobb angle was not statistically significant. The result of the interactions between the Lenke classification, the batch type, and the Cobb angle evolution was insignificant. The result of the interactions between the initial Risser score category, the batch type, and the Cobb angle evolution was negligible. The results returned by the ANOVA analysis showed that only belonging to the study group was a factor with a statistically relevant impact on the development of the Cobb angle.

The analysis of the interaction between vitamin D3 deficiency after one year, the type of physical therapy, and the evolution of the Cobb angle was performed to see if patients who had vitamin D3 deficiency initially and after one year had more severe changes in the Cobb angle, concerning the type of physiotherapy performed. The results returned by the ANOVA analysis showed that the interaction between vitamin D3 deficiency and the kind of physiotherapy performed did not correlate in a statistically significant way in the evolution of the Cobb angle, nor did the type of therapy taken as an individual factor. However, not correcting vitamin D3 deficiency worsened the Cobb angle in a statistically significant way.

It is observed in Table 6 that the serum calcium level was similar in both groups between the initial moment and one year after the application of the treatment, while the values of vitamin D and melatonin increased.

## 4. Discussion

The treatment of childhood and adolescent idiopathic scoliosis follows the guidelines based on Cobb angle and other factors such as age, gender, or menarche status for girls. The staging of treatment includes physiotherapy, wearing a brace, and surgery for advanced cases. None of these steps takes into account the possible existence of a metabolic factor that aggravates the evolution of the disease.

The Introduction states that these novel therapies significantly impact patients’ quality of life and medical costs. Although this is an attractive perspective, a protocol for this kind of triple therapy is yet to be defined. Few studies present those three agents as medication with a positive impact on idiopathic scoliosis outcomes. However, these agents are treated separately in some studies. More recently, this type of approach has been changing, and some authors propose dual or even triple therapy. Calcium, melatonin, and vitamin D can be administered altogether. Doses and routes of administration are not precise. In addition, the frequency of administration and the proper patient age and disease stage to start this therapy are yet to be discussed. 

Based on this perspective, in our novel study we tried to verify, based on the hypothesis, the connection between the serum level of 25-OH-vitamin D, calcium, and melatonin and the curve progression (Cobb angle), in patients who still have growth remaining representing this fact through a Risser score between 0 and 3. In the treatment of a patient with idiopathic scoliosis, many factors can influence the result. Our study sought to verify as many combinations as possible so that the result is not at risk of bias. Thus, our results follow the application of standard treatment with the addition of vitamin D, calcium, and melatonin in patients in the study group to see their evolution compared to the control group.

The two groups were balanced and comparable; otherwise, we would not have been able to compare the averages of the final Cobb angles between the two groups, and we would not have been able to rely on the effectiveness of the treatment.

Our results showed progression of the Cobb angle in the control group. The evolution of the mean Cobb angle was significantly different for patients who received vitamin D3, calcium, and melatonin supplements compared to those who did not. Moreover, a slight decrease in the mean evolution of the Cobb angle could be observed in patients that received supplements.

Melatonin is a hormone primarily released by the pineal gland at night. Melatonin synchronizes circadian rhythms in invertebrates, including sleep–wake timing and blood pressure regulation.

Azzedine et al., after a systematic search in different databases until July 2021, showed that the concentration and rhythm of melatonin secretion could play an essential role in influencing the pathogenesis of adolescent idiopathic scoliosis [28].

In a study with level III control evidence, Goultidis et al. show that higher melatonin levels were observed in conservatively treated patients with AIS. The study also indicates that melatonin deficiency was not associated with AIS progression [29].

However, Bordner et al. show that scoliosis seen in chickens after pinealectomy resembles adolescent idiopathic scoliosis in humans. It has been suggested that pineal deficiency in the hormone melatonin is responsible for this phenomenon in both species [30]. 

Regarding the mechanism of action of melatonin in idiopathic scoliosis, Azeddine et al. showed a genetic cause of adolescent idiopathic scoliosis (AIS). More importantly, the findings can be used in classifying patients with AIS [28]. 

Concerning melatonin’s action on the skeleton, Amstrup et al. showed that melatonin may affect bone metabolism through bone anabolic and antiresorptive effects. An age-related decrease in peak melatonin levels at nighttime is well documented, which may increase bone resorption and bone loss in the elderly.

Furthermore, melatonin improves bone formation by promoting the differentiation of human mesenchymal stem cells (hMSCs) into the osteoblastic cell lineage [31].

Finally, Grivas et al. stated that, in the last decade, melatonin has been used as a therapeutic chemical in a large spectrum of diseases, mainly in sleep disturbances and tumors. Melatonin may play a role in the pathogenesis of scoliosis (neuroendocrine hypothesis) but, to date, the data available cannot support this hypothesis. Uncertainties and doubts still surround the role of melatonin in human physiology and pathophysiology, and future research is needed [32].

In our study, the melatonin mean was 8.44 pg/mL for the control group, which is at the bottom level of the normal range (8–16 pg/mL), and 7.66 pg/mL for the study group. The value increased to 14 pg/mL after one-year supplementation treatment with melatonin and also correlated with a positive outcome for the patients enrolled. In our opinion, patients should be monitored for their melatonin levels and supplemented if values are close to the baseline or below.

Regarding vitamin D, Beling et al. investigated whether vitamin D-deficient AIS patients experienced higher pain before or immediately after posterior spine fusion (PSF) surgery. The results showed no differences in preoperative SRS-30 score, pre- and postoperative central curve angles, or estimated blood loss across vitamin D groups. In addition, trajectories of NRS indicated no differences in pain during the first 72 h after surgery [33]. However, Herdea et al. showed that the positive correlation between vitamin D and calcium and the negative correlation with the Cobb angle are proof that patients with idiopathic scoliosis should be investigated regularly for these pathologies [26]. Furthermore, Ng et al. postulate that vitamin D deficiency and insufficiency affect AIS development by affecting the regulation of fibrosis, postural control, and BMD. Subclinical deficiency of vitamin K2, a fat-soluble vitamin, is also prevalent in adolescents; therefore, the high prevalence of vitamin D deficiency may be related to decreased fat intake. The authors suggested that further studies are required to elucidate the possible role of vitamin D in the pathogenesis and clinical management of AIS [34]. 

Our study saw an increase in the Cobb angle if vitamin D deficiency was not treated accordingly. A positive outcome was shown by patients that managed to increase their 25-OH-vitamin D level. From our results, it is necessary to evaluate the blood sample levels of 25-OH-vitamin D every six months and treat those deficient or insufficient with 2000 UI per day. Further studies could verify a threshold that patients should achieve and maintain regarding the 25-OH-vitamin D level.

Regarding the association with calcium, Goździalska et al. performed a cross-sectional study of two groups of patients with scoliosis and an age-matched control group. The groups, including patients with adolescent idiopathic scoliosis (AIS) and the control group, were divided into premenarcheal and postmenarcheal girls. The results showed that the phosphate–calcium balance and PTH level seemed normal in patients with AIS. However, the calcitonin level in girls with AIS was 2-fold lower than in healthy subjects. The deficiency of vitamin D may be involved in AIS [35]. Finally, in an extensive cohort study Lam et al. addressed whether bone health improvement from the initial 2-year Ca+Vit-D supplementation could persist as subjects approached peak bone mass at 6 years, i.e., after 4 years of supplement discontinuation. The results showed that after 4-year supplement discontinuation, the treatment effect from the initial 2-year supplementation mostly dissipated, indicating the need for continued supplementation in AIS girls to sustain therapeutic improvement of bone health as subjects approach peak bone mass [27].

In our study, the calcium level was irrelevant to the evolution of the disease and remained constant even after one year of supplementation with 600 mg of calcium per day. In our opinion, calcium should be taken along with vitamin D due to the transporter mechanism of vitamin D, which brings calcium to bones, thus increasing bone density. As other authors suggested, some patients who have scoliosis tend to have osteopenia. Further studies could check bone mass density at the first evaluation of a scoliosis patient and then follow up regularly. Calcium levels should be monitored only to ensure that the patient is within the range of the average value after supplementation.

Steffan et al. showed that, regarding physical therapy, these methods could be applied during inpatient or outpatient treatment or intensified in the practice of specialized therapists. As well as Schroth and Vojta therapy, Moore’s physical therapy is at present the most common and effective method in the biological treatment of idiopathic scoliosis [23].

Physiotherapy represents a factor that could balance our results. In our study, Schroth therapy brought some improvement, but the impact of Schroth treatment on the final result does not differ significantly from the impact of classical physical therapy. This result allowed the comparison of the impact of Schroth therapy on the final result vs. the effects of the classical treatment, as the two groups of children were sufficiently homogeneous. This result showed that, concerning the available sample, the impact of Schroth therapy on the final result did not vary considerably from the effects of classical physical treatment [25].

This study is also accompanied by some minus points, such as the small study group of participants, which makes a powerful analysis difficult. Additionally, an unequal number between the two sexes and the short follow-up period are to be noted. 

A future direction of research is to study a larger sample of patients and to consider whether vitamin D3 deficiency in younger children leads to an aggravation of scoliosis faster than in older patients. Our results did not exclude a link between the patient’s age and the evolution of the Cobb angle but only indicated that, compared to the available data and regarding the division into two age groups using the median value, the two age groups did not show statistically significant differences. The fact that the mean evolution of the Cobb angle for the older age group was lower than that for the younger age category is explained by the fact that the worsening of scoliosis was lower as the growth process was nearing completion. In addition, another future direction of research would be to test whether vitamin D3 deficiency correlates with an unfavorable evolution of the patients over a certain threshold of the Cobb angle.

In our study, participants were regularly asked for signs, symptoms, and side effects generated by the treatment with vitamin D, calcium, or melatonin. None of our patients experienced any side effects due to the daily administration of these drugs.

Overall, some idiopathic scoliosis cases occur and develop based on insufficient levels of vitamin D, calcium, and melatonin and must be treated as such. Scoliosis is divided into two categories: idiopathic at 90% (without another secondary disease) and secondary at 10% (these are rarer cases, especially for those with neuromuscular disorders, tetraparesis, etc.). Our study indicated 25-OH-vitamin D as a valid marker that can be further checked for all idiopathic scoliosis patients. This study suggests that among idiopathic cases, a (quite significant) percentage may have a detectable cause. It may be proven that from the large group of idiopathic scoliosis patients, some have a substrate of vitamin D + calcium + melatonin deficiency. We cannot estimate the percentage of such cases. However, in a future paper, we will address this issue.

## 5. Conclusions

Our preliminary results show that low vitamin D levels can predict a significant increase in Cobb angle. 

Patients with idiopathic scoliosis may benefit from a novel treatment by supplementation with vitamin D, calcium, and melatonin, where there is a serum level deficiency of 25-OH-vitamin D (below 30 ng/mL). 

Some idiopathic scoliosis cases occur and progress due to insufficient vitamin D, calcium, and melatonin levels.

## Figures and Tables

**Figure 1 children-09-00758-f001:**
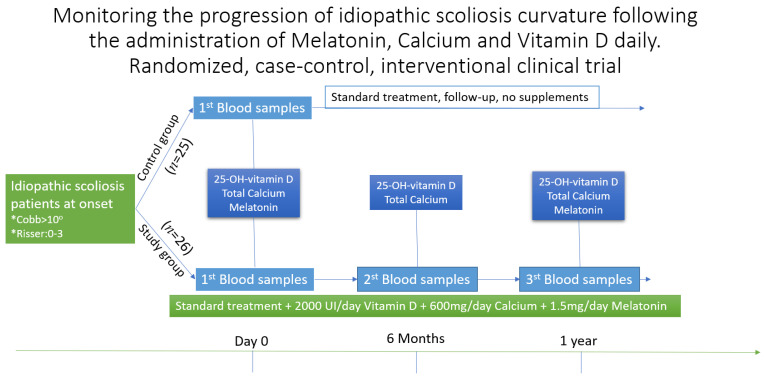
Follow-up and analysis table for patients in the study “Monitoring the progression of idiopathic scoliosis curvature following the administration of Melatonin 1.5 mg/daily, Calcium 600 mg/daily and Vitamin D 2000 UI/daily. The randomized, case–control, interventional clinical trial”. Initially, when enrolling in the study, both groups did all three tests, then, at six months and one year, only the study group repeated the tests. After six months, the study group repeated only the blood samples for vitamin D and calcium, being only a stage verification. * Other requirements for inclusion were Cobb angle of more then 10^0^ and a Risser score between 0 and 3.

**Figure 2 children-09-00758-f002:**
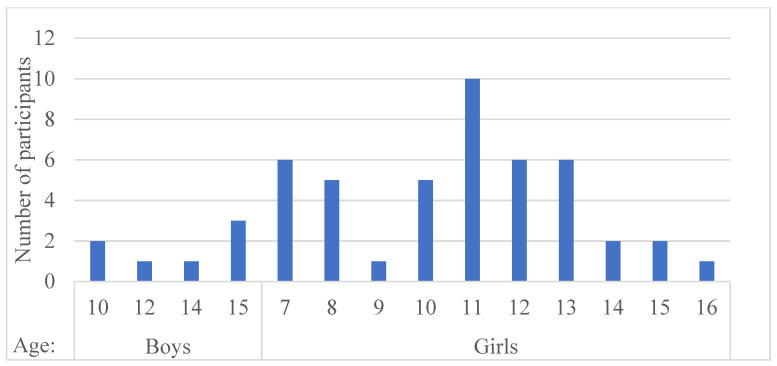
Distribution of patients by age and sex. There were many female patients, a predominant feature of the disease.

**Table 1 children-09-00758-t001:** Distribution of patients in the study and control groups according to age and Cobb angle at study entry.

	Study Group		Control Group	
	Age (Years)	Cobb Angle	Age (Years)	Cobb Angle
Total	26 patients		25 patients	
Average	11	23.73°	11.2	27°
SD	2.35	10.82°	2.61	13.91°
Max	15	50°	16	54°
Min	7	10°	7	10°
95% CI	[10.1, 11.9]	[19.8°, 28.1°]	[10.2, 12.2]	[22.2°, 33.1°]

**Table 2 children-09-00758-t002:** Analysis of the Cobb angle between the study group and the control group upon initial starting point, after one year of treatment, and its evolution between these two intervals.

	Control Group (*n* = 25)	Study Group (*n* = 26)	F-Test	*T*-Test
Initial Cobb angle	27°	23.73°	*p* = 0.13	*p* = 0.35
Cobb angle after 1 year	32.16°	22.76°	*p* = 0.01	*p* = 0.005
Cobb angle evolution after 1 year	+5.16 °	−0.97°	*p* = 6.81 × 10^−6^	*p* = 0.24

**Table 3 children-09-00758-t003:** Analysis of the Cobb angle evolution for patients who initially had vitamin D3 deficiency grouped by their final serum levels, after one year.

	Cobb Angle Evolution	F-Test	*T*-Test
Low level of vitamin D initially and also finally (*n* = 25)	+4.04°	*p* = 0.35	*p* = 6.85 × 10^−5^
Low level of vitamin D initially and high level finally (*n* = 21)	−1.42°		

**Table 4 children-09-00758-t004:** Analysis of the Cobb angle based on each physiotherapy type, comparing initial values and results after one year.

	Classical Physiotherapy (*n* = 21)	Schroth Physiotherapy (*n* = 30)	F-Test	*T*-Test
Initial Cobb angle	25.42°	25.26°	*p* = 0.12	*p* = 0.96
Cobb angle at one year	29.04°	26.2°	*p* = 0.10	*p* = 0.45

**Table 5 children-09-00758-t005:** Analysis of the Cobb angle by age groups (from 7–11 years old and 12–16 years old) at the starting point, after one year of treatment, and its evolution.

	7–11 Years Old Age Group	12–16 Years Old Age Group	F-Test	*T*-Test
Initial Cobb angle	20.41°	31.81°	*p* = 0.48	*p* = 0.0004
Cobb angle evolution after 1 year	+2.62°	+1.27°	*p* = 0.009	*p* = 0.40
Cobb angle after one year	23.03°	33.09°	*p* = 0.32	*p* = 0.002

**Table 6 children-09-00758-t006:** Comparison between the level of blood tests taken between the initial moment and 1 year after the applied treatment.

	Control Group	Study Group
Initial Cobb angle	27.00°	23.73°
Cobb angle at one year	32.16°	22.76°
Initial vit. D3	19.39 ng/mL	19.93 ng/mL
Vit. D3 at one year	-	39.88 ng/mL
Initial Ca	9.87 mg/dL	9.87 mg/dL
Ca at one year	-	9.90 mg/dL
Initial melatonin	8.44 pg/mL	7.66 pg/mL
Melatonin at one year	-	14.00 pg/mL

## Data Availability

All data are registered at “Grigore Alexandrescu” Children’s Emergency Hospital, Bucharest, Romania.

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
