# Peer review of "Controlling the Progression of Curvature in Children and Adolescent Idiopathic Scoliosis Following the Administration of Melatonin, Calcium, and Vitamin D"

_children, 2022, doi:10.3390/children9050758_

Round 1
Reviewer 1 Report
In this study, the authors analyzed that the impact of using Melatonin, Calcium, and Vitamin D, respectively, on idiopathic scoliosis patients. Results showed that these drugs positively affected the illness progression quantified by the spine curvature. On the bases of these facts, they concluded that overall treatment of patients with idiopathic scoliosis is improved by supplementation with Vitamin D, Calcium, and Melatonin.
I evaluate this is valuable information for the physicians managing idiopathic scoliosis patients.
The methodology of this study was precisely explained and acceptable. Also, the limitations of this project were well indicated.
Therefore, I think this manuscript is appropriate for publication in Children journal. However, I have a couple of minor requests to be considered as stated below. After they have been resolved, I will judge this manuscript can be accepted and published by the Children journal.
*I ask the authors that correction parts will be shown in red color in the revised manuscript.
- description error?
Page 4 line 163
Did 51 cases of this study meet not only inclusion but also exclusion criteria?
Is this information correct? I guess the authors selected the patient that fell into inclusion criteria only.
Page 6 line 195 “… we could apply”
Is this part the end of sentence or the midstream of sentence?
Page 10, line 394 “… last decade, ‘s Melatonin …”
This part should be revised properly.
Author Response
In this study, the authors analyzed that the impact of using Melatonin, Calcium, and Vitamin D, respectively, on idiopathic scoliosis patients. Results showed that these drugs positively affected the illness progression quantified by the spine curvature. On the bases of these facts, they concluded that overall treatment of patients with idiopathic scoliosis is improved by supplementation with Vitamin D, Calcium, and Melatonin.
I evaluate this is valuable information for the physicians managing idiopathic scoliosis patients.
The methodology of this study was precisely explained and acceptable. Also, the limitations of this project were well indicated.
Therefore, I think this manuscript is appropriate for publication in Children journal. However, I have a couple of minor requests to be considered as stated below. After they have been resolved, I will judge this manuscript can be accepted and published by the Children journal.
*I ask the authors that correction parts will be shown in red color in the revised manuscript.
Comment 1. description error?
Page 4 line 163
Did 51 cases of this study meet not only inclusion but also exclusion criteria?
Is this information correct? I guess the authors selected the patient that fell into inclusion criteria only.
Response 1:: Dear reviewer, thanks for your review. We have updated the phrase to "the required criteria" line 171 to be more clear about the patients enrolled in the study.
Comment 2 Page 6 line 195 “… we could apply”
Is this part the end of sentence or the midstream of sentence?
Response 2: The initial phrase is "This test was necessary to decide what kind of test we could apply." line 199 - This sentence refers to the F test being applied to understand better the correct statistical test that can be used applied further. Thus, the T-test was applied next by correlating the results from F-test. I hope that this is the answer that you were looking for. If not, please guide us to the required phrase.
Comment 3: Page 10, line 394 “… last decade, ‘s Melatonin ….”
This part should be appropriately revised.
Response 3: The type-o was corrected. line 406 now
Thank you for reviewing our article!
Reviewer 2 Report
General comments
The manuscript is an original article reporting the results from a prospective randomized case‑control study conducted at the Pediatric Orthopedics Department at "Grigore Alexandrescu" Children's Emergency Hospital, Bucharest, Romania. The article is well written, and the therapeutic approach proposed in this study is interesting.
A few comments:
Cobb angle is a relevant follow up criteria. Did the authors also collect functional data ? It would have been of interest to include these criteria in the evaluation.
Is there side effects for melatonin administration ? Have the participants been asked for these during the follow‑up ? That can maybe be discussed in the discussion section.
Will the author consider extending the follow‑up after one year ?
Remarks on Figures and Tables
Figure 1: maybe another term than “cases” for the group with treatment might be used ? please mention the follow‑up for this cases group. What about physiotherapy ?
Would it be possible to add the number of participants in each group to the Figure ?
Figure 2: Although it is nice to see the distribution, I can’t understand this figure. Please add labels for axis. Is there two boys of ten yo or ten boys of two yo?
Table 2: there is a line finishing in the middle of the figure (below “age”)?
Here are a few minor remarks:
43: in humans, and confirmed
59: rather “protective role”: can you further develop
69: which role ?
92: how was the randomization performed ? Who assigned the odd and even numbers to the participants ?
149: “T‑test, F‑test were used”
157: please add a point at the end of the sentence
160: MELA is defined there but the word metanonin is frequently used before; either define MELA earlier or simply use “melatonin” in the whole article?
165: sample size instead of sample zize
176: would it be possible to add the units (25/26 patients, degrees for the angles etc.)?
189-190: the sentence is unclear
Please use a uniform word for F‑test and T‑test (or F test and T test), sometimes you use the hyphen and capital, sometimes not
338: instead of “novel study” would you consider using “current study”?
394: there is a “’s” to remove
481: “Low vitamin D levels can predict a significant increase in Cobb angle.” not sure you can conclude that with this study design
485: please temper a bit by adding “Our data suggest that…”
502: “Acknowledgments: None” Wouldn’t you acknowledge the families who accepted to participate ?
Author Response
General comments
The manuscript is an original article reporting the results from a prospective randomized case‑control study conducted at the Pediatric Orthopedics Department at "Grigore Alexandrescu" Children's Emergency Hospital, Bucharest, Romania. The article is well written, and the therapeutic approach proposed in this Study is interesting.
A few comments:
Comment 1Cobb angle is a relevant follow up criteria. Did the authors also collect functional data ? It would have been of interest to include these criteria in the evaluation.
Response1: Thank you for you question. What do you mean functional data? Do you mean if the patients had pain or movement limitation? In our cases, none of these were reported.
Comment 2 Is there side effects for melatonin administration ? Have the participants been asked for these during the follow‑up ? That can maybe be discussed in the discussion section.
Response 2 : In our evaluation, we regularly asked for signs, symptoms, and side effects generated by any of Vitamin D, Calcium, or Melatonin. None of our patients experienced any side effects due to the daily administration of these drugs. Therefore, we added in lines 489-491.
Comment 3 Will the author consider extending the follow‑up after one year ?
Response 3 : the patients are followed-up until they reach 18 years of age. In this article, we present the results after at least one year of follow-up.
Remarks on Figures and Tables
Comment 4 Figure 1: maybe another term than "cases" for the group with treatment might be used ? please mention the follow‑up for this cases group. What about physiotherapy ?
Would it be possible to add the number of participants in each group to the Figure ?
Response 4 : We changed cases for the Study, and we specified the number of patients in each group, as seen in Figure 1, line 132.
Physiotherapy was described in line 141-142 "Patients underwent either conventional physical therapy or Schrot therapy and were monitored every six months for follow-up."
Follow up period is mentioned in the Figure 1, as having at least 1 year of follow up and explained in line 155.
Comment 5 Figure 2: Although it is nice to see the distribution, I can't understand this figure. Please add labels for axis. Is there two boys of ten yo or ten boys of two yo?
Response 5 : in order for the reader to better understand the structure of our patients, we mentioned Gender and Age on one side and the number of participants on the other. We modified and added "Age:" and "Number of participants" in line 176 to better reflect our age distribution.
Comment 6 Table 2: there is a line finishing in the middle of the figure (below "age")?
Response 6 : it was repaired.
Here are a few minor remarks:
Comment 7 43: in humans, and confirmed
Response 7 : modified in line 43
Comment 8 59: rather "protective role": can you further develop
Response 8 : we added "slowing down evolution " in line 60 to reflect the protective role better.
Comment9: 69: which role ?
Response 9: This is our hypothesis that those 3 drugs may play in the onset and development of idiopathic scoliosis
Comment 10 92: how was the randomization performed ? Who assigned the odd and even numbers to the participants ?
Response 10 : randomization was presented in lines 101-104. While adding patients into the database, they had an updated ID every time a new patient was in. Every odd-numbered patient went for the study group, while the even-numbered patient went for the control group.
Comment 11 149: "T‑test, F‑test were used"
Response 11 : updated in line 157
Comment 12 157: please add a point at the end of the sentence
Response 12 : done
Comment 13 160: MELA is defined there but the word metanonin is frequently used before; either define MELA earlier or simply use "melatonin" in the whole article?
Response 13 : we added MELA due to the laboratory settings and tag name using Melatonin (MELA) - if anyone would like to check the LCMS method for this blood sample. We are using Melatonin in the article.
Comment 14 165: sample size instead of sample zize
Response 14 : done
Comment 15 176: would it be possible to add the units (25/26 patients, degrees for the angles etc.)?
Response 15 : updated in Table 1
Comment 16 189-190: the sentence is unclear
Response 16 : We reffer that we applied statistical methods for the results at 1 year timeframe.
Comment 17 Please use a uniform word for F‑test and T‑test (or F test and T test), sometimes you use the hyphen and capital, sometimes not
Response 17 : done
Comment 18 338: instead of "novel study" would you consider using "current study"?
Response 18 : the other reviewers suggested to show that this is a Novel study and to use this term. Originaly, it was "in our study".
Comment 19 394: there is a "’s” to remove
Response 19 : done
Comment 20 481: “Low vitamin D levels can predict a significant increase in Cobb angle.” not sure you can conclude that with this study design
Response 20 : Due to the small sample size, a regression analysis of any sort wouldn't be reliable, despite the fact that it could bring stronger results(causality). The t-test can indeed only reveal an association and not a causality relation between the low vitamin D level and a higher Cobb angle, but using a one-tail t-test we showed that the statistical hypothesis tested saying that children with low vitamin D levels who don't correct the deficit have, on average, a higher Cobb angle compared to those who corrected the deficit within 12 months, is verified with 95% confidence level.Moreover, we performed an interaction analysis, between 3 variables: deficit of vitamin D after 1 year, the type of physical therapy(classical/Schrot), and the evolution of the Cobb angle. We started with contingency tables and we verified the preliminary conclusions drawn with an ANOVA analysis. The latter showed that while the interaction between the first two variables does not significantly impact the evolution of the Cobb angle, the deficit of vitamin D does have such an effect. We conclude by saying that the association between a low level of vitamin D and the evolution of the cobb Angle is irrefutable. While we cannot appropriately quantify it within the current sample(with odds ratio, for example), the direction of the association is statistically proven.
Comment 21 485: please temper a bit by adding “Our data suggest that…”
Response 21 : modified in line 496
Comment 22 502: “Acknowledgments: None” Wouldn’t you acknowledge the families who accepted to participate ?
Response 22 : we added "We acknowledge the families who accepted to participate." in line 528
Dear reviewer, thank you for your comments and remarks.
Reviewer 3 Report
The manuscript is a prospective, randomized, case-control, interventional study aimed to analyze the impact of using Melatonin, Calcium, and Vitamin D, respectively, on idiopathic scoliosis patients. Results show that these drugs positively affected the illness progression quantified by the spine curvature. Overall treatment of patients with idiopathic scoliosis is improved by supplementation with Vitamin D, Calcium, and Melatonin.
I read the article with interest, the title is well thought out and faithfully reflects the content of the study, although it would be appropriate to specify the characteristics of the study to be clearer to the reader
The abstract is adequately developed, and it is useful to frame the purpose of the study.
In the introduction, the characteristics of Idiopathic scoliosis have been shortly described. The discussion is sufficiently developed, even if a little too synthetic.
Nevertheless, some minor changes are needed to be considered suitable for publication.
Comment 1: It would be appropriate to apply it, adding more information and dividing it into paragraphs (background, material and methods, results and conclusion) to make it clearer to the reader the characteristics and the design of the study.
Comment 2: In the abstract: It would be appropriate to refer to the fact that it is a prospective, randomized, case-control, interventional study, it does not seem to be very clear.
Comment 3: In the introduction: “The lack of vertebral rotation with lateral deviation denotes scoliotic attitude, a benign condition that remits in a supine position.” Please add suitable bibliographic references.
Comment 4: In the introduction: “Deficiency is problematic in both children - rickets - and adults - osteomalacia - and correlates with an increased incidence of fractures due to bone fragility and the prospect of associated muscle damage” Could you add appropriate bibliographical references?
Comment 5: In the introduction: further features on diagnosis and treatment should be briefly described inserting some references, for example (Di Maria F, et al. (2021) "Immediate Effects of Sforzesco® Bracing on Respiratory Function in Adolescents with Idiopathic Scoliosis. Healthcare (Basel))".
Comment 7: In the discussion: It would be appropriate to describe what could be done in future studies to improve these preliminary data.
Comment 8: Finally, additional English editing is needed. The Non-Native Speakers of English Editing Certificate was not signed.
Author Response
Comment 1: It would be appropriate to apply it, adding more information and dividing it into paragraphs (background, material and methods, results and conclusion) to make it clearer to the reader the characteristics and the design of the study.
Response1: Abstract was rewritten.
Comment 2: In the abstract: It would be appropriate to refer to the fact that it is a prospective, randomized, case-control, interventional study, it does not seem to be very clear.
Response2: Dear reviewer, thank you for your suggestion. It was added in the abstract in lines 18-19.
Comment 3: In the introduction: “The lack of vertebral rotation with lateral deviation denotes scoliotic attitude, a benign condition that remits in a supine position.” Please add suitable bibliographic references.
Response3: Dear reviewer, thank you for your suggestion. Reference was added in line 30.
Comment 4: In the introduction: “Deficiency is problematic in both children - rickets - and adults - osteomalacia - and correlates with an increased incidence of fractures due to bone fragility and the prospect of associated muscle damage” Could you add appropriate bibliographical references?
Response4: Dear reviewer, thank you for your suggestion. A reference was added in line 55.
Comment 5: In the introduction: further features on diagnosis and treatment should be briefly described inserting some references, for example (Di Maria F, et al. (2021) "Immediate Effects of Sforzesco® Bracing on Respiratory Function in Adolescents with Idiopathic Scoliosis. Healthcare (Basel))".
Response 5: Dear reviewer, we have added such useful information in lines 68-69.
Comment 6: In the discussion: It would be appropriate to describe what could be done in future studies to improve these preliminary data.
Response 6: Dear reviewer, we described what further studies could do in lines 480-488. If it is not sufficient, we can expand the discussion.
Comment 7: Finally, additional English editing is needed. The Non-Native Speakers of English Editing Certificate was not signed.
Response 7: the English language has been edited.
Dear reviewer, thank you for your comments .
Reviewer 4 Report
Dear Authors,
This is an interesting study. Unfortunately, the paper is not written well. Therefore, many parts should be re-written (especially the results). Statistical strategy should be rearranged based on the aim (written in the discussion) and with the claim in the conclusion. Furthermore, for the same reasons, the methods should be reconsidered as well.
Introduction gives a good overview of the background and of the aim of the study.
The methods present some inconsistencies and there are some parts that were not shown in the methods but were done in the results (ex. the use of ANOVA). The results are mixed with the discussion and they are written in a way which are difficult to understand.
Discussion is confusing as well. The aim in this section is different from the introduction.
Conclusion does not match with the statistical strategy and the results. I think there are not the elements to claim that a low vitamin D level can predict a significant increase in Cobb angle. To answer to this question another study and statistical strategies should be performed.
Study design: mono-center, prospective, randomized, case-control, interventional study.
Aim: verify whether the administration of Melatonin, Vitamin D, and Calcium supplements to patients diagnosed with idiopathic scoliosis (IS) will slow down or stop the progression of the curves.
Population: IS pts. ,with Risser 0-3, >10° Cobb angle, 7-16 yo., at least 1 y. follow-up.
Intervention: treatment with Melatonin, Vitamin D and Calcium.
Collection: demographic (age, sex), clinical and radiographic data (Lenke, Cobb, Risser), blood samples results each 6 months for intervention group (serum Melatonin, Vitamin D and Calcium) and administration supplements.
Outcome: not completely understandable.
Suggestions
Introduction
Line 28-29: please explain what you mean.
Line 62-66: SRS and SOSORT guidelines suggest different. I would not say it is conventional.
Line 74-75: please put a reference to show that the interest is increasing.
Line 77: please put reference to show that in developing countries scoliosis has remarkable frequency (I assume you mean that in developing countries the frequency is higher?).
Methods
The protocol is pretty clear. No sample size or power analysis?
Line 88: treating and surgically treating. Please if explain more in details the other types of treatment as well (or just write treatment if you cover all the different types of treatment).
Line 96-98: please explain the phrase. Did you include only patients that were prospectively diagnosed or also patients that were previously as well?
Line 99: please explain how you randomized the patients. Did the patient and the physician know of the treatment or not? Or you mean randomized to show that the patient was anonymized?
Line 138: please explain which type of brace (or braces if more than one type) was used.
Line 148-152: please use always the same verb tense in order to be consistent.
Line 154-163: please explain better what statistical strategy you used. In the conclusion you claim that low vitamin D can predict significant increase of Cobb angle. Nevertheless, if this is the aim, the statistical strategy is not correct. For treatment prediction a linear regression (or other similar regression types which might be even better) should be performed. T-test is not sufficient.
Results
Results are very confusing. Results and discussion sections are often mixed and the data are shown in a way that is difficult to objectively understand. Tables and figures are good and explain better the results than the text.
Line 176-181: please rephrase it.
Line 208-210: this should be in the discussion section.
Line 219-221: please rephrase it.
Line 281-283: this should be in the discussion section.
Line 300-301: was it done or it will be done later in the future? In that case it should be in the discussion as well.
Line 313: here you talk about ANOVA but in the methods it was not written.
Discussion
Line 344-352: what do you mean with this paragraph? Why this part is in red?
Line 353-356: the aim was to see if the administration of the three drugs would have helped the patients with idiopathic scoliosis. Here you state that the aim was to test the connection between serum levels of Vitamin D, Melatonin, calcium and curve progression. The aim in the discussion is different and in this case another strategy should have been chosen.
Author Response
Introduction
Comment 1Line 28-29: please explain what you mean.
Response 1: in introduction, we intended to explain the differences between scoliosis and scoliotic attitude (or functional scoliosis).
Comment 2:Line 62-66: SRS and SOSORT guidelines suggest different. I would not say it is conventional.
Response 2: we removed the word "conventional" in line 62
Comment 3 Line 74-75: please put a reference to show that the interest is increasing.
Response 3: we added a reference in line 77
Comment 4 :Line 77: please put reference to show that in developing countries scoliosis has remarkable frequency (I assume you mean that in developing countries the frequency is higher?).
Response 4: we modified "has a remarkable frequency" to "it is also present".
Methods
Comment 5 The protocol is pretty clear. No sample size or power analysis?
Response 5 : We talked about sample size/power analysis in the discussions on lines 475-485.
Comment 6 Line 88: treating and surgically treating. Please if explain more in details the other types of treatment as well (or just write treatment if you cover all the different types of treatment).
Response 6 : we modified and deleted "and surgically treating". We cover all types of treatment.
Comment 7 Line 96-98: please explain the phrase. Did you include only patients that were prospectively diagnosed or also patients that were previously as well?
Response 7 : new patients and previously diagnosed patients with scoliosis were included if they met the inclusion criteria.
Comment 8 Line 99: please explain how you randomized the patients. Did the patient and the physician know of the treatment or not? Or you mean randomized to show that the patient was anonymized?
Response 8 : while gathering our patients, they were included in the study group or control group based on they're assignment number (register numbers assigned upon addition like 1,2,3, etc). Meaning that every odd-numbered patient went for the study group, while the even-numbered patient went for the control group. The study was blinded to the patients but not for the physician.
Comment 9 Line 138: please explain which type of brace (or braces if more than one type) was used.
Response 9 : We added "Cheneau brace" in line 140.
Comment 10 Line 148-152: please use always the same verb tense in order to be consistent.
Response 10 : Lines 150-154 were updated.
Comment 11 Line 154-163: please explain better what statistical strategy you used. In the conclusion you claim that low vitamin D can predict significant increase of Cobb angle. Nevertheless, if this is the aim, the statistical strategy is not correct. For treatment prediction a linear regression (or other similar regression types which might be even better) should be performed. T-test is not sufficient.
Response 11 : Due to the small sample size, a regression analysis of any sort wouldn't be reliable, despite the fact that it could bring stronger results(causality). The t-test can indeed only reveal an association and not a causality relation between the low vitamin D level and a higher Cobb angle, but using a one-tail t-test we showed that the statistical hypothesis tested saying that children with low vitamin D levels who don't correct the deficit have, on average, a higher Cobb angle compared to those who corrected the deficit within 12 months, is verified with 95% confidence level.
t-Test: Two-Sample Assuming Equal Variances |
|
|
|
|
|
|
Low VitD3 initially and after 12 months |
Low VitD3 initially and normal level after 12 months |
Mean * |
4,04 |
-1,428571429 |
Variance |
21,04 |
17,75714286 |
Observations |
25 |
21 |
Pooled Variance |
19,54779221 |
|
Hypothesized Mean Difference |
0 |
|
Df |
44 |
|
t Stat |
4,178551585 |
|
P(T<=t) one-tail |
6,84845E-05 |
|
t Critical one-tail |
1,680229977 |
|
P(T<=t) two-tail |
0,000136969 |
|
t Critical two-tail |
2,015367574 |
|
*Average of Cobb angle differences between the initial value and the value after 12 months
We corroborate this result by performing a t-test to assess if there is a significant difference between the evolution of the Cobb angle for those who had classical physical therapy as opposed to Schrot physical therapy, and we did not obtain a statistically significant difference between the two samples.
t-Test: Two-Sample Assuming Equal Variances |
||
|
|
|
|
Classical physical therapy |
Schrot therapy |
Mean* |
29,04761905 |
26,2 |
Variance |
227,547619 |
136,5103448 |
Observations |
21 |
30 |
Pooled Variance |
173,6684159 |
|
Hypothesized Mean Difference |
0 |
|
Df |
49 |
|
t Stat |
0,759463059 |
|
P(T<=t) one-tail |
0,225606991 |
|
t Critical one-tail |
1,676550893 |
|
P(T<=t) two-tail |
0,451213981 |
|
t Critical two-tail |
2,009575237 |
|
*Average of Cobb angle differences between the initial value and the value after 12 months
Moreover, we performed an interaction analysis between 3 variables: deficit of vitamin D after 1 year, the type of physical therapy(classical/Schrot), and the evolution of the Cobb angle. We started with contingency tables, and we verified the preliminary conclusions drawn with an ANOVA analysis. The latter showed that while the interaction between the first two variables does not significantly impact the evolution of the Cobb angle, the deficit of vitamin D does have such an effect.
deficitVitD3final evolCobb |
yes 4.040000 |
no -1.428571 |
Schrot deficitVitD3final yes no |
yes 3.090909 -1.25 |
no 4.785714 -2.00 |
Df Sum Sq Mean Sq F value Pr(>F) |
deficitVitD3final 1 341.3 341.3 17.060 0.000169 *** |
schrot 1 5.8 5.8 0.288 0.594154 |
deficitVitD3final:schrot 1 14.1 14.1 0.703 0.406448 |
Residuals 42 840.3 20.0 |
--- |
Signif. codes: 0 ‘***’ 0.001 ‘**’ 0.01 ‘*’ 0.05 ‘.’ 0.1 ‘ ’ 1 |
We conclude by saying that the association between a low level of vitamin D and the evolution of the cobb Angle is irrefutable, and while we cannot appropriately quantify it within the current sample(with odds ratio, for example), the direction of the association is statistically proven.
Results
Results are very confusing. Results and discussion sections are often mixed and the data are shown in a way that is difficult to objectively understand. Tables and figures are good and explain better the results than the text.
Commment 12 Line 176-181: please rephrase it.
Response 12 : Lines were rephrased in line 179.
Comment 13 Line 208-210: this should be in the discussion section.
Response 13 : were moved in lines 357-359.
Comment 14 Line 219-221: please rephrase it.
Resp[onse 14 : it was rephrased in lines 217-218.
Comment 15 Line 281-283: this should be in the discussion section.
Response 15 : were moved in lines 482-485
Comment 16 Line 300-301: was it done or it will be done later in the future? In that case it should be in the discussion as well.
Response 16 : the discussion followed in the next couple of phrases. We changed "will further analyze" into "analyzed" in lines 293.
Comment 17 Line 313: here you talk about ANOVA but in the methods it was not written.
Response 17 : we added ANOVA in methods at line 157.
Discussion
Comment 18 Line 344-352: what do you mean with this paragraph? Why this part is in red?
Response 18 : we were requested by the editor to provide more info about out study.
Comment 19 Line 353-356: the aim was to see if the administration of the three drugs would have helped the patients with idiopathic scoliosis. Here you state that the aim was to test the connection between serum levels of Vitamin D, Melatonin, calcium and curve progression. The aim in the discussion is different and in this case another strategy should have been chosen.
Response 19: To prove a connection between the administration of Calcium, Vitamin D and Melatonin with the Scoliosis disease, both aspects must be considered. For example, when you administer Vitamin D, you must know the serum base values for the given patient at the mb beginning and then at follow-ups. Of course, these values must be correlated with the progression of the disease in order to quantify any efficiency of these drugs. This means that one cannot administer Vitamin D, Calcium, and Melatonin to any given patient before having the base serum levels. It may be possible that a given patient has average values, meaning that it does not need supplementation. However, our article proved that the administration of Vitamin D, Calcium, and Melatonin to idiopathic scoliosis patients with low serum levels would benefit. As y read in our last part of the article, mostly 25-OH-vitamin D is the only serum level needed to be tested if one could not also test Calcium and Melatonin, having a direct correlation with the evolution of the disease.
Thanks for reviewing our article!
Round 2
Reviewer 4 Report
Dear Authors,
I find this study very interesting. I think that this topic should be studied more and probably it will in the future.
The paper improved and is a bit more clear. Nevertheless, the result section is still unclear and methods, results and discussion are mixed. Therefore, I would suggest to make it more concise and show the results only without explaining if the data is good, confirmed the aim(s). I do think that this should be in the discussion section only since it is interpretation of the data.
The English language should need some revision as well, since some grammatical and logical errors are present. More concise phrases might help on the comprehension of the manuscript.
Furthermore, there still some inconsistencies:
Line 173: here a power analysis was made.
Line 476: it is stated that a post hoc power analysis was not performed.
What is the difference between the power analysis performed I the results and the statement in the discussion. Furthermore, if something is in the results it should be stated in the methods as well.
Conclusion
In your study the aim was to see if vitamin D, calcium and melatonin treatment would have helped patients with IS. The concept behind was that treating these patients in this way the outcome (reduction of Cobb angle) would have been better. With your results you can state that there is a correlation between Vitamin D assumption and Cob stabilization/reduction. Therefore the treatment can be helpful on treating these patients. Nevertheless, the mean of Vitamin D in both control and action group are within the normal range. Therefore, with these data you can only see a correlation between a treatment (vitamin D, Calcium and Melatonin assumption) and another one (normal treatment with no Vitamin D, calcium and Melatonin treatment). The idea that low levels of vitamin D will have a negative effect on Cobb angle cannot be stated:
- To state this the strategy of the study should have been different with two groups one low levels of vitamin D and the other one with normal levels.
Author Response
Comment 1: I find this study very interesting. I think that this topic should be studied more and probably it will in the future.
The paper improved and is a bit more clear. Nevertheless, the result section is still unclear and methods, results and discussion are mixed. Therefore, I would suggest to make it more concise and show the results only without explaining if the data is good, confirmed the aim(s). I do think that this should be in the discussion section only since it is interpretation of the data.
Response 1: Result section was modified accordingly (lines 184-198, 201-207, 210-217, 220-255, 258-260), and Discussion section was updated (lines 288-307, 418-425, 443-447). We hope that the results section now provides more precise information.
Comment 2: The English language should need some revision as well, since some grammatical and logical errors are present. More concise phrases might help on the comprehension of the manuscript.
Response2: the English language was revised.
Comment 3: Furthermore, there are still some inconsistencies:
Line 173: here a power analysis was made.
Response3: power analysis was moved in the Methods section in line 166. We hope that everything is ok now.
Comment4:Line 476: it is stated that a post hoc power analysis was not performed.
Response4: Lines 472-479 and reference 38 were removed
Comment5: What is the difference between the power analysis performed I the results and the statement in the discussion. Furthermore, if something is in the results it should be stated in the methods as well.
Response5: we moved the power analysis and removed the unnecessary lines as requested.
Conclusion
Comment 6: In your study the aim was to see if vitamin D, calcium and melatonin treatment would have helped patients with IS. The concept behind was that treating these patients in this way the outcome (reduction of Cobb angle) would have been better. With your results you can state that there is a correlation between Vitamin D assumption and Cob stabilization/reduction. Therefore the treatment can be helpful on treating these patients. Nevertheless, the mean of Vitamin D in both control and action group are within the normal range. Therefore, with these data you can only see a correlation between a treatment (vitamin D, Calcium and Melatonin assumption) and another one (normal treatment with no Vitamin D, calcium and Melatonin treatment). The idea that low levels of vitamin D will have a negative effect on Cobb angle cannot be stated:
To state this the strategy of the study should have been different with two groups one low levels of vitamin D and the other one with normal levels.
Response6 :We presented in Table 6 the mean values for vitamin D for both the control group(19,39 ng/mL) and the study group(initial 19,93 ng/mL; after one year 39,88). We notice both groups have initial insufficient vitamin D mean levels( less than 20 ng/mL; the normal levels considered are above 30 ng/mL), but the mean level for the study group falls into the normal range after one year. The control group had vitamin D levels between 10 and 42 with only 3 normal values measured initially. The study group had vitamin D levels between 9 and 41 with also only 3 normal values measured initially, while after 1 year the vitamin D values were between 24 and 48, with 22 normal values, out of the total of 26.
Given this particular structure of the 2 groups, after eliminating the exceptions(3 in the case of the control group and 4 in the case of the study group) we can characterize the control group as "low initial values -low final values of vitamin D" and the study group as "low initial values-normal final values of vitamin D". What we have proven in our study is that low levels of vitamin D which have been corrected by administrating vitamin D supplements lead to a better evolution of the Cobb angle as opposed to low levels of vitamin D which stay low during the course of one year. It would have been impossible to compare two groups, one with low initial vitamin D levels and the other one with normal initial vitamin D levels as no such data was available. In our experience, the vast majority of children with idiopathic scoliosis also suffer from low levels of vitamin D, the exceptions being very rare. So the conclusion we want to underline, which is supported by the statistical analysis provided within our study, is that correcting the low levels of vitamin D is a key element in having a favorable evolution of the Cobb angle while treating idiopathic scoliosis with either classical or Schrot therapy.
Thank you for your comments.